# Anatomical Predictions using Subject-Specific Medical Data

**Marianne Rakic** [1]                                             MRAKIC@MIT.EDU
[1] *CSAIL, MIT*
**John Guttag** [1]                                               GUTTAG@MIT.EDU
**Adrian V. Dalca** [1,2]                                         ADALCA@MIT.EDU
[2] *MGH, HMS*

## Abstract

Changes over time in brain anatomy can provide important insight for treatment design or scientific analyses. We present a method that predicts how a brain MRI for an individual will change over time. We model changes using a diffeomorphic deformation field that we predict using function using convolutional neural networks. Given a predicted deformation field, a baseline scan can be warped to give a prediction of the brain scan at a future time. We demonstrate the method using the ADNI cohort, and analyze how performance is affected by model variants and the subject-specific information provided. We show that the model provides good predictions and that external clinical data can improve predictions.

## 1. Introduction

Changes in neuroanatomy, such as brain development or neurodegeneration, are important indicators of overall health and clinical trajectory. We present a learning-based method to predict future brain anatomy from a single *baseline* MRI scan. Our method can also incorporate other clinical data; such as age, gender, and genetic information.

Longitudinal brain scan datasets have typically been used to extract correlation between brain structures and biological markers or clinical data (Biffi et al., 2010; Potkin et al., 2009; Risacher et al., 2010; Shen et al., 2010). However, providing an accurate prediction of the entire brain can give a richer phenotype for use in analysis or clinical practice. Models have been used to simulate brain evolution, taking as input one or more baseline scans. These have generally employed simple linear predictive models and had limited success (Blanc et al., 2012; Fleishman et al., 2015; Modat et al., 2014; Dalca et al., 2015). We focus on predicting the evolution of brain anatomy, using one previous scan along with external data. A recent machine learning model directly predicts future images using a black box CNN approach, without characterizing the anatomically meaningful changes (Ravi et al., 2019).

We model changes as deformations between the baseline scan and a follow-up scan, building on learning-based diffeomorphic registration methods (Balakrishnan et al., 2019; Dalca et al., 2019; de Vos et al., 2017; Ashburner, 2007; Dalca et al., 2019; Hernandez et al., 2009; Yang et al., 2017). We design a neural network that predicts such deformations and present initial results using the *ADNI II* dataset (Mueller et al., 2005a).

## 2. Methods

**Model.** Let $\boldsymbol{x}_0$ be the baseline subject brain scan, and $\boldsymbol{a}_0$ be a vector of subject-specific medical *attributes*, such as age, diagnosis, or genetic information. We predict the brain scan $\boldsymbol{x}_t$, at a later time $t$. We assume that evolution is captured by a deformation field $\boldsymbol{\phi}_v^{(t)}$ via $\boldsymbol{x}_0 \circ \boldsymbol{\phi}_v^{(t)}$ where $\circ$ represents the spatial warp operation, and we obtain a noisy observation

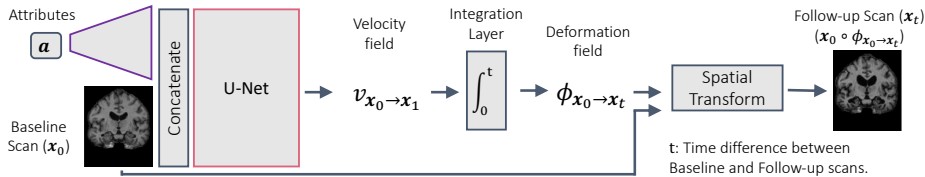

Figure 1: **Architecture.** The concatenated input scan and decoded attributes are inputs to a U-Net (Ronneberger et al., 2015), which estimates the velocity field of anatomical changes. This is then used to predict a future scan, and the predictions are compared to the true changes for longitudinal training examples to provide a loss.

$\boldsymbol{x}_t$ via the likelihood: $p(\boldsymbol{x}_t|\boldsymbol{\phi}_v^{(t)};\boldsymbol{x}_0,\boldsymbol{a}_0) = \mathcal{N}(\boldsymbol{x}_t;\boldsymbol{x}_0\circ\boldsymbol{\phi}_v^{(t)},\sigma^2\mathbb{I})$, where $\mathcal{N}(\cdot;\mu,\Sigma)$ is the normal distribution with mean $\mu$ and covariance $\Sigma$, and $\sigma^2$ accounts for image noise.

We parametrize the deformations $\boldsymbol{\phi}_v^{(t)}$ using a stationary velocity field, $v$. To encourage the predicted deformation field to be anatomically plausible, we employ a smoothness prior for the velocity field. Let $\boldsymbol{u}_v$ be the displacement field such that $\boldsymbol{\phi}_v = Id + \boldsymbol{u}_v$. Also, we let

$$p(\boldsymbol{\phi}_v;\boldsymbol{x}_0,\boldsymbol{a}_0) \propto \exp\{-\gamma\,\|\nabla\boldsymbol{u}_{v_0}\|^2\}, \tag{1}$$

where $\gamma$ is a parameter that regulates the importance of the priors, $\nabla$ is the spatial differential operator, and $v_0$ indicates that the velocity field is a function of $\boldsymbol{x}_0$ and $\boldsymbol{a}_0$.

The complete data likelihood is then written as:

$$p_\theta(\boldsymbol{x}_t;\boldsymbol{x}_0,\boldsymbol{a}_0) = \int p(\boldsymbol{x}_t|\boldsymbol{\phi}_v^{(t)};\boldsymbol{x}_0,\boldsymbol{a}_0)p(\boldsymbol{\phi}_v^{(t)};\boldsymbol{x}_0,\boldsymbol{a}_0)d\boldsymbol{\phi}_v^{(t)}. \tag{2}$$

**Learning.** Because equation (2) is intractable, we use a point estimate for $\hat{\boldsymbol{\phi}}_v^{(t)}$, and maximize $p(\boldsymbol{x}_t;\boldsymbol{x}_0,\boldsymbol{a}_0) \approx p(\boldsymbol{x}_t|\hat{\boldsymbol{\phi}}_v^{(t)};\boldsymbol{x}_0,\boldsymbol{a}_0)$. To obtain this point estimate, we approximate $v$ using a neural network $g_\theta(\boldsymbol{x}_0,\boldsymbol{a}_0)$, shown Figure 1. It takes as input a baseline scan and optional clinical attributes, and outputs a velocity field, $v$. The network parameters, $\theta$, are learned using stochastic gradient algorithms applied to a training dataset of longitudinal observations. Given a new pair of baseline and follow-up images $\boldsymbol{x}_0,\boldsymbol{x}_t$, we find optimal parameters $\theta$ by maximizing the posterior $\log p(\boldsymbol{\phi}_v^{(t)}|\boldsymbol{x}_t;\boldsymbol{x}_0,\boldsymbol{a}_0)$. For each sample $\{\boldsymbol{x}_t,\boldsymbol{x}_0\}$ and predicted velocity field $v$, we use the loss

$$\mathcal{L}(\theta;\boldsymbol{\phi}_v^{(t)},\boldsymbol{x}_t,\boldsymbol{x}_0,\boldsymbol{a}_0) = -\log p(\boldsymbol{\phi}_v^{(t)}|\boldsymbol{x}_t;\boldsymbol{x}_0,\boldsymbol{a}_0) = \frac{1}{2\sigma^2}\left\|\boldsymbol{x}_t - \boldsymbol{x}_0\circ\boldsymbol{\phi}_v^{(t)}\right\|^2 + \gamma\,\|\nabla\boldsymbol{u}_v\|^2 + cst.$$

**Inference.** To predict future scans, given learned parameters $\theta^*$, we use the likelihood:

$$\boldsymbol{x}_t^* = \arg\max_{\boldsymbol{x}_t} p_{\theta^*}(\boldsymbol{x}_t;\boldsymbol{x}_0,\boldsymbol{a}_0), \approx \arg\max_{\boldsymbol{x}_t} p_{\theta^*}(\boldsymbol{x}_t|\boldsymbol{\phi}_v^{(t)};\boldsymbol{x}_0,\boldsymbol{a}_0),$$

following (2) and using a point estimate. We first compute $\boldsymbol{\phi}_v^{(t)} = \int_0^t v\,dt'$ for $v = g_{\theta^*}(\boldsymbol{x}_0,\boldsymbol{a}_0)$, and then compute $\hat{\boldsymbol{x}}_t = \boldsymbol{x}_0\circ\boldsymbol{\phi}_v^{(t)}$.

## 3. Experiments

**Data.** We use the ADNI dataset (Mueller et al., 2005b) as pre-processed in (Dalca et al., 2018). For medical attributes, we select features often used in analysis: years of education, sex, APOE3, APOE4, diagnosis of the patient, and results of a Mini-Mental State Examination. Segmentation maps obtained via FreeSurfer including 29 anatomical structures are used in evaluating registration results using the Dice score and surface distance between the ground truth follow-up maps and the propagated segmentation labels.

**Comparative Methods.** We consider three comparative methods. The first assumes no anatomical change between the baseline and follow-up scan. The second one gives the

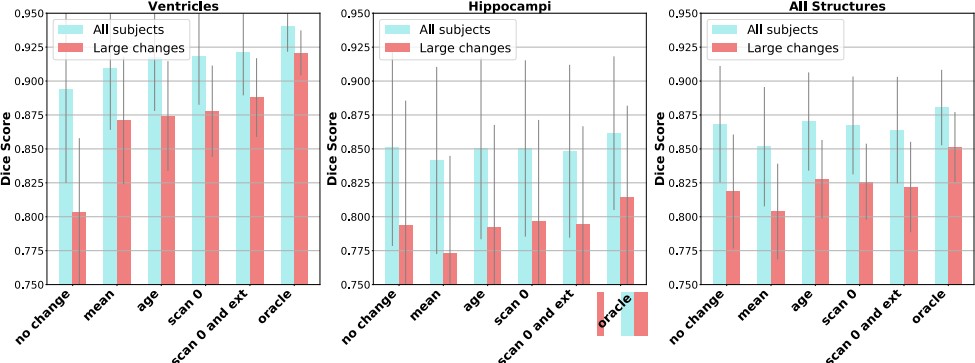

Figure 2: **Dice score evaluated for the test set.** The name of the model corresponds to the input of the network, with `ext` being the full set of external data.

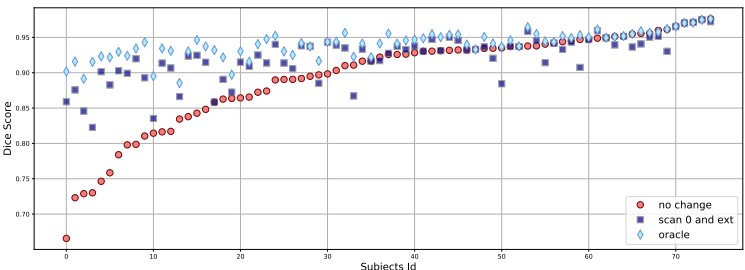

Figure 3: **Relative change of performance for the different subjects evaluated on ventricles.** The subjects are ordered by decreasing change of ventricle volume between the two scans.

deformation field obtained by integration of the average registration velocity field (mean) in the training set (Ashburner, 2007). The third (oracle) is an upper bound that uses the follow-up scan and outputs the deformation field by registering the baseline **and** the follow-up scan (Balakrishnan et al., 2018).

**Model Variants.** We train variants of the proposed model, with results given in Figure 2 and Figure 3. We distinguish between subjects who experience large changes between the baseline and follow-up scans and the ones that don't. We observe that the proposed model is able to significantly improve on the baselines in some structures, such as the ventricles, and come close to the upper bound. Specifically, for ventricles, adding clinical information tends to further improve the results. For other structures not affected by external data, such as the hippocampi, we hypothesize that external information is instead captured in the medical scan itself and extracted by our network. Surface distance gives similar results.

## 4. Conclusion

We propose a deformation model and neural network architecture for predicting anatomical changes from a single baselines scan. In initial experiments, we show that the proposed architecture can extract meaningful information and lead to promising predictions. We limit our model to shape changes to focus on modelling neurodegeneration and atrophy. It would further be interesting to determine whether predictions could be enhanced by leveraging label maps during training and by incorporating intensity changes in the model.

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
