# OpenReview forum: "Anatomical Predictions using Subject-Specific Medical Data"
_MIDL.io/2020/Conference — MIDL 2020_

### Official Review · AnonReviewer3 · 2020-03-09
**interesting paper with rather abstract description**

**Rating:** 3
**Confidence:** 4

**Review:**

This paper covers an interesting topic in predicting the evolution (i.e. decline) of human brains for neurodegenerative disease. This is nice work, but the motivation is not well spelled out. It would be beneficial to clearer state that it might be useful to tell at a given point how a disease will develop. In the future it might be even possible to give the well known "brain age" by analyzing development curves. This clinical aspect is not well developed in the paper, but would greatly strengthen the significance of the described work.

To me, it was unclear how the input data looks like. Did you use follow up brain scans from the ADNI study? If yes, what was the time between baseline and follow up? I also didn't get your comment on considering "three baselines". Do you mean you trained three different networks? Does the curve "no change" in fig. 3 mean that these are the dice coefficients of the original images?
Again, I don'T think the term "three baselines" is well chosen. There should be one baseline and one follow-up scan.

The results and the considerations drawn on them has to be more clear. I do not see the conclusion that additional clinical information improves the results being backed by presented data.

---

### Official Review · AnonReviewer2 · 2020-03-10
**A nice paper on brain image prediction**

**Rating:** 4
**Confidence:** 4

**Review:**

This short paper proposed a method for predict future brain images with the use of deformation filed. The proposed method is interesting and focuses on an important problem. It is generally a nice work and some comments are as following:
Pros:
- The proposed method is technically valid.
- This idea is promising and will likely be hot in the field of brain image prediction.

Cons:
- What are the clinical values of the proposed method?
- What are the differences between this paper and Dalca et al 2019 (nips)?

---

### Official Review · AnonReviewer4 · 2020-03-12
**An interesting way to incoporate subject-specific demographic and medical information into brain image classification problem**

**Rating:** 3
**Confidence:** 4

**Review:**

The paper presented a way to incorporate subject-specific demographic and medical information along with the brain image scan to improve the deep neural network for predicting longitudinal brain deformation. The results demonstrated a level of effectiveness of the proposed approach, although I have some specific concerns and comments that listed in the detailed review shown below.

*Major comments*

• The author didn't include the time difference between the training and predicted image in the "vector of subject-specific medical attributes". This raise my first question:
	• It is not clear that, for multi-timepoint longitudinal data, whether and how how did the authors accounted for registration between different length of timepoints.

• Structural segmentation are used for generating evaluation metric (dice score and surface-distance). It's not clear how the segmentation is generated.
	• If it's generated using image registration, then that means the registration-derived deformation field are considered as ground truth. Then why not using this deformation field directly for validation?
	• Furthermore, if the the deformation field is the output to be optimized, should the in deformation field itself be used to calculate the lost function, and image similarity (the current term of the loss function) been used  as validation metric? This way, no additional registration-based segmentation is compulsory during the evaluation
	• This lead to my third point for critical question regarding the ultimate purpose of the study. In the introduction, the author claimed that "providing an accurate prediction of the entire brain can give a richer phenotype for use in analysis or clinical practice." It would be better to state/discuss more about what would be the ultimate end goal of this generative model.
		○ If segmentation is mainly used for generating clinical-relevant features such as volume or shape, would it make more sense to predict the longitudinal structural change (e.g. volume, shape, thickness, etc)? What's the comparative advantage of predicting the brain deformation, given that there are significant more parameters to train and fine-tune, which will make the generative model less robust. (E.g. Would the benefit be something like the ability to perform voxel-based morphological analysis?)
		○ Similarly, and furthermore,  if better clinical score/diagnosis is the ultimate purpose, would this method provide better/comparable/worse results in predicting future clinical diagnosis, as compared to alternative generative models that predict clinical scores/diagnosis directly, given that the latter need even less parameters to trained with?


*Minor comments*
• Some of the equations might need to be clarified a little bit more
	• In the paragraph before equation (2), $\theta_v = Id + u_v$ : what is $I$ and $d$?
	• In equation (4), it is not clear what does cst mean?
•
• I would suggest to add a little bit more details about how the  "subject attributes" are concatenated to the baseline scan images, either in the text or in the Figure 1 legend. Are those attributes remapped to the size of the image and concatenated as additional channels (although the schematic diagram makes it seems like the images has been vectorized before concatenating into 1-dimensional tensor)? If so what happened if some subject doesn't have specific attributes recorded (such as the education level or clinical score)?

• Validation:
	• Figure 2 and legend is a bit confusing.
		○ What does scan 0 mean?
		○ Shall *Ext* be referred to as the full set of "subject-specific medical attributes" rather than "external data" as stated in the legend?
		○ Why the upper bound (ground truth) name "oracle"
The baseline of "integration of the average registration velocity ﬁeld (mean) in the training set" seems a strange baseline metrix to me

---

### Official Review · AnonReviewer1 · 2020-03-13
**Interesting approach to longitudinal prediction of brain imaging**

**Rating:** 4
**Confidence:** 3

**Review:**

The authors propose a data-driven method to predict the anatomical changes in the brain over time, encoded as a deformation field. The main contribution is to employ a neural network to predict the parameters f a diffeomorphic deformation field that morphs the images between exams, instead of directly predicting the anatomy at a later exam.
The paper addresses an important challenge in neuroimaging and is well written and validated

It is unclear if the method is predicting a deformation over time, this is , its evolution so that the intermediate deformed shapes between the initial scan and the follow on are meaningful or realistic, or if the method predicts the anatomy at follow up exam and uses  a diffeomorphic field  to regularise the problem or to try to give some intuition to what the neural network is doing.  Is the time between both imaging procedures always the same?

Authors indicate that there are some optional attributes (encoded in vector $a$) that can be used, but it is not clear if this is actually used in the experiments, and what is the impact. How is this reflected in Fig 2 or Fig 3?

Authors are say that they use the segmentations to train the model but Fig. 1 shows images ath both ends -Ithink this could be further clarified.

---

### Meta-Review · Area_Chair1 · 2020-04-06
**MetaReview of Paper236 by AreaChair1**

**Rating:** 4

**Metareview:**

All reviewers have given a positive evaluation to the paper (2 strong accept, 2 weak accept). Therefore,  I  strongly recommend its acceptance.

**Paper Type:**

both

---

### Decision · Program_Chairs · 2020-04-11

Accept